# Characterization and Biocompatibility Assessment of Boron Nitride Magnesium Nanocomposites for Orthopedic Applications

**DOI:** 10.3390/bioengineering10070757

**Published:** 2023-06-25

**Authors:** Mary S. Jia, Shelby Hash, Wendy Reynoso, Mostafa Elsaadany, Hamdy Ibrahim

**Affiliations:** 1Department of Biomedical Engineering, University of Arkansas, Fayetteville, AR 72701, USA; maryjia@uark.edu (M.S.J.); mselsaad@uark.edu (M.E.); 2Department of Mechanical Engineering, University of Tennessee, Chattanooga, TN 37403, USA; xbx252@mocs.utc.edu (S.H.); mkw818@mocs.utc.edu (W.R.)

**Keywords:** nanocomposites, corrosion, biocompatibility, orthopedics, magnesium

## Abstract

Magnesium (Mg) has been intensively studied as a promising alternative material to inert metallic alloys for orthopedic fixation devices due to its biodegradable nature inside the body and its favorable biocompatibility. However, the low mechanical strength and rapid corrosion of Mg in physiological environments represent the main challenges for the development of Mg-based devices for orthopedic applications. A possible solution to these limitations is the incorporation of a small content of biocompatible nanoparticles into the Mg matrix to increase strength and possibly corrosion resistance of the resulting nanocomposites. In this work, the effect of adding boron nitride (BN) nanoparticles (0.5 and 1.5 vol.%) on the mechanical properties, corrosion behavior, and biocompatibility of Mg-based nanocomposites was investigated. The properties of the nanocomposites fabricated using powder metallurgy methods were assessed using microstructure analyses, microhardness, compression tests, in vitro corrosion, contact angle, and cytotoxicity tests. A significant increase in the microhardness, strength, and corrosion rates of Mg–BN nanocomposites was detected compared with those of pure Mg (0% BN). Crystalline surface post-corrosion byproducts were detected and identified via SEM, EDX, and XRD. Biocompatibility assessments showed that the incorporation of BN nanoparticles had no significant impact on the cytotoxicity of Mg and samples were hydrophilic based on the contact angle results. These results confirm that the addition of BN nanoparticles to the Mg matrix can increase strength and corrosion resistance without influencing cytotoxicity in vitro. Further investigation into the chemical behavior of nanocomposites in physiological environments is needed to determine the potential impact of corrosive byproducts. Surface treatments and formulation methods that would increase the viability of these materials in vivo are also needed.

## 1. Introduction

Magnesium (Mg), Mg-based alloys, and Mg-based composites represent a new class of biodegradable biomaterials that have been investigated for various orthopedic applications such as skeletal fixation hardware. The interest in Mg, compared with currently more prevalent inert metals such as stainless steels, titanium alloys, and cobalt–chromium alloys, is due to its biodegradable nature inside the body, allowing it to degrade after the completion of the bone healing process [1,2,3]. This is expected to eliminate problems associated with currently available inert metals-based fixation devices due to their permanent existence in the body. Some of the most common problems are stress shielding, possible infection, and future fracture of the fixation device, which may require removal through a second revision surgery [1,2,4,5]. In addition, Mg has preferable physical, mechanical, and osteogenic properties compared with other biodegradable materials [3,6,7,8,9,10]. For instance, Mg is a lightweight metal with an elastic modulus and compressive yield strength closer to that for bone in comparison to other metallic implants, which eliminates the possibility of stress shielding from weakening existing bone [7,11]. Stress shielding is the alteration of the mechanical environment of bone when proximal bone–implant interfaces induce adaptive bone remodeling, catalyzing bone loss [12,13,14]. Due to disparities in the mechanical properties of bone compared with common orthopedic implant metals such as titanium, biomaterials with a high similarity in physical behavior, such as Mg, are attractive to prevent bone loss induced by the stress shielding phenomenon. Mg also has a well-characterized degradation behavior that eventually results in a complete dissolution of the corrosion products in the biological environment [15]. The major limitation of Mg, however, is its insufficient mechanical strength and fast corrosion (degradation) rates in physiological environments, causing poor biomechanical performance, especially for load-bearing applications. This may result in a loss of a Mg-based implant’s mechanical integrity before sufficient tissue regeneration and osteolysis, which can be associated with excessive hydrogen gas accumulation [7,16].

Alternative biodegradable metals, such an iron (Fe) and zinc (Zn), are also candidates for the development of biodegradable orthopedic implants. Fe orthopedic implant materials as an abundant metal are associated with bone homeostasis. Additionally, despite having a higher strength, Fe has a higher elastic modulus compared with Mg and Zn and a slower degradation rate, which runs a higher risk of stress shielding with a corrosion behavior that is more like that of permanent implants [17,18,19]. Further, Zn, which can stimulate bone mineralization and growth, has an ideal corrosion resistance causing moderate degradation rates. However, Zn has a low modulus of elasticity and strength compared with Mg and Fe, which presents difficulty for its use in biomedical applications [20,21,22]. Polymer-based biodegradable materials such as the resorbable materials poly(l- or d,l-lactic acid), poly(glycolic acid), and polycaprolactones (PCL) have also been of interest for orthopedic applications and have been FDA-approved as implant materials [23,24]. The predominant mechanisms of degradation in these polymers are generally hydrolytic or enzymatic with constituents ranging from wholly synthetic to biologically derived materials. The degradation rate and material properties are modular in polymers and are, therefore, easier to engineer. However, the primary barriers in generating polymer orthopedic implants are the resulting intermediates, acidic byproducts of degradation, and wear debris, causing inflammation or other undesirable effects [5,25,26]. The toxicity of Mg biomaterials is comparatively less severe, and the elastic modulus is of sufficient similarity to that of bone [27,28]. Therefore, the main obstacles in creating magnesium-based orthopedic implant materials are the modulation of quick degradation and an increase in material strength.

Several strategies have been employed to overcome the limitations of Mg, such as surface modifications [29], alloying [30], and heat treatments. Recently, Mg composites reinforced with nanoparticles have shown promising increases in strength and ductility without significant weight penalties while maintaining a high cytocompatibility in vitro [31,32,33]. Notably, Mg nanocomposites composed of rare earth oxide nanoparticles have been investigated due to their excellent strength and corrosion resistance [33,34]. However, the incorporation of boron in orthopedic biomaterials has been of recent interest due to the physiological role of boron as a micronutrient with an essential role in osteogenesis and bone maintenance [15]. The incorporation of boron into implanted biomaterials such as titanium stimulates a pro-angiogenic effect in wound healing by upregulating angiogenesis markers, upregulating growth factors and cytokines, and facilitating pre-osteoblast differentiation by activating genes associated with osteogenesis [35,36]. A similar effect can be observed in BN, with BN nanotubes promoting osteogenic differentiation in vitro, and implants coated with BN had enhanced bone fracture healing in mouse models [37,38]. It has also been previously demonstrated that Mg nanocomposites fabricated with low BN percentages using powder metallurgy demonstrated high levels of grain refinement, and enhanced localized recrystallization compared with pure Mg [39,40,41]. As a result of the expected microstructural characteristics of Mg–BN nanocomposites imparted via BN addition to Mg, we hypothesize that Mg–BN nanocomposites will have the improved mechanical properties and increased corrosion resistance needed for orthopedic implant applications. Despite the well-investigated characterization of the degradation and biocompatibility of Mg in solvents like physiological environments, similar characterizations of Mg–BN nanocomposites in a physiological environment have yet to be sufficiently studied. To the best of the authors’ knowledge, no studies in the literature reported the effect of the addition of boron nitride (BN) to a Mg matrix on the corrosion properties and biocompatibility in a physiological-like solvent for bone implant applications. 

To this end, the objective of this study is to investigate the mechanical properties, corrosivity, and cytotoxicity of Mg nanocomposites reinforced with boron nitride (BN) nanoparticles (0.5 and 1.5 vol.% BN) for orthopedic fixation device applications. Microhardness and compression tests were performed to investigate the effect of the BN nanoparticles on the mechanical properties. In vitro electrochemical corrosion tests were performed to investigate the degradation and corrosion mechanism of the prepared nanocomposites. Scanning electron microscopy (SEM) and energy dispersive X-ray spectroscopy (EDX) were additionally used to characterize and visualize the surface of each experimental group. These data collectively assess the viability of Mg–BN nanocomposites as orthopedic implants and inform needed improvements.

## 2. Materials and Methods

### 2.1. Synthesis of Mg–BN Nanocomposites

BN nanoparticles, 50 nm in diameter, were incorporated in Mg powder to synthesize BN–Mg nanocomposites of Mg–0.5%BN and Mg–1.5%BN vol.% through powder metallurgy and microwave-assisted rapid sintering methods [39,42]. Mg powder with >98.5% purity and 60–300 µm particle size (Merck, Darm-stadt, Germany) was used as the base metal. The BN nanoparticles provided by Sigma Aldrich were used as the reinforcement. The mechanically mixed powders were then cold compacted uniaxially at 1000 psi before being sintered at 630 °C in a 2.45 GHz, 900 W Sharp microwave oven. The resulting material was then deformed through hot extrusion at 350 °C at a 20 to 1 ratio, producing 8 mm rods that were finally machined into 7.2 mm diameter and 3 mm thick coupons for their characterizations [39]. In a similar fashion, 0%BN Mg (pure Mg) was synthesized using this method to generate a control for mechanical testing. For cytotoxicity studies, high-purity Mg (ultra-pure Mg: 99.9% purity) as-rolled coupons (Goodfellow, Pittsburgh, PA, USA) were also used as a control group in the in vitro cytotoxicity test due to the known biocompatibility of high-purity Mg [30]. In this paper, “pure Mg” refers to the Mg–0%BN samples prepared using the powder metallurgy method used to fabricate the nanocomposite samples, and “ultra-pure Mg” is used to refer to the control group used in the in vitro cytotoxicity/immersion test only.

### 2.2. Mechanical Testing

The microhardness test was conducted using a Shimadzu HMV-G21s Micro Vickers hardness tester (Kyoto, Japan). To prepare the samples for the microhardness test, the three sample types, Mg–0%BN, Mg–0.5%BN, and Mg–1.5%BN, were solidified in a polymer base using a Buehler Simplimet II mounting press (Lake Bluff, IL, USA). The nanocomposite samples solidified in a polymeric base were then polished using 100- to 3000-grit SiC sandpaper. The samples were then tested on a Shimadzu HMV-G21s Vickers microhardness tester at ambient temperature using a test load of 9.807 N, holding for 15 s, and indents were imaged at 10× magnification. Ten indents were performed on each of the tested groups. Additionally, cylindrical compression test specimens, prepared according to the ASTM E9-09 standard, were tested using an Instron-5569 universal testing machine. The test was conducted at 50% relative humidity and 20 °C.

### 2.3. Porosity Tests

The density of the prepared samples was determined based on the Archimedes’ principle, using an MXBAOHENG MH-300A density meter. The measured density was then compared with the theoretical density and percentage porosity was calculated for each case. The theoretical density was determined based on the rule of mixture.

### 2.4. Contact Angle

Due to the importance of hydrophilicity in the interaction between orthopedic implants and surrounding osteoblasts and osteoclasts, the wettability of the coupons was determined by measuring the static contact angle three times on each side of the Mg–0%BN, Mg–0.5%BN, and Mg–1.5%BN coupon groups with four coupons of each sample using the sessile drop technique with a CAM PLUS MICRO contact angle meter (Cheminstruments, Fairfield, OH, USA) at ambient temperature. A micrometer syringe was used to dispense a droplet of 2–3 µL of distilled water or simulated body fluid (SBF) onto the surface of each coupon type. Measurements were replicated for water and SBF sanding samples immediately before analysis to obtain a more accurate contact angle. The resulting contact angle measured through projecting the droplet image on a protractor and an average contact angle was calculated for six different measurements of three similarly sized droplets on each side of the coupons, wiping the coupons clean with a Kim wipe after each measurement [15,43,44,45].

### 2.5. Electrochemical Corrosion Test

The corrosion behavior of the prepared Mg nanocomposites was determined by conducting electrochemical corrosion tests in SBF. The SBF was optimized to contain ion concentrations approximating that of blood plasma and a fresh SBF was prepared for each respective corrosion test [43]. Prior to each test, the nanocomposites were sequentially polished with 400–2000 grit SiC sandpapers, washed with ethanol, and dried. The electrochemical tests (potentiodynamic polarization tests) were performed using a three-electrode system with the nanocomposite sample serving as the working electrode, while graphite and saturated calomel (SCE) were used as the counter and reference electrode, respectively. The three-electrode system was submerged in the SBF for 600 s, allowing the system to stabilize before potentiodynamic polarization (PDP) tests were performed. PDP tests were conducted using a Gamry potentiostat Interface 1010E model (Gamry Instruments, Warminster, PA, USA) at room temperature. The connection between the outer wire and the sample was coated with a nonconductive epoxy so that only the desired surface area of the top face of the sample was exposed to the SBF during the test. After polishing the exposed surfaces of Mg nanocomposites with the 400–2000 grit SiC sandpapers, the epoxy was cured. Prior to the first test, the potentiostat was connected and calibrated [46]. The three-electrode system was assembled and placed in a 150 mL beaker of SBF [47]. The potentiostat was connected to the electrodes by attaching five terminals to the electrodes and the ground, ensuring that the only surface the metal of the banana clips touched was the electrodes. Tafel curves were generated using the DC corrosion method through the Gamry instruments framework software and they were used to determine the corrosion current densities (i_corr_). The potentiostat was programmed to send a DC potential varying from −0.25 to 0.25 V relative to the measured open circuit potential (OCV) through the electrodes at a scan rate of 1 mV/s while monitoring the current between electrodes that formed anodic and cathodic curves [48]. From the x-axis of the graph generated by the potentiostat, the corrosion current density i_corr_ (A/cm^2^) was calculated by dividing each x-axis value by the tested area of the sample (cm^2^). The i_corr_ value was obtained from the graph using Tafel extrapolation and linear fitting approach [49]. The corrosion current densities were used to study the effect of adding the reinforcing phases on the PDP corrosion characteristics, as an indication for the corrosion rates. The corrosion rate was assessed based on the corrosion potential and current density values i_corr_ [50,51].

### 2.6. Cytotoxicity and In Vitro Immersion Test

Sterilization protocols were performed according to the recommended sterilization of small Mg parts for cell culture experiments [30]. The Mg coupon samples were individually sterilized in ultrasonically pulsed ethanol for 3 min, acetone for 5 min, and isopropanol for 20 min. The coupons were then immersed in deionized H_2_O for three sonication steps at five minutes each. Prior to cell culture, the samples were immersed in 70% ethanol for 5 min and sterilized in UV light for 10 min on each side [51]. The resulting samples were washed with sterile 1× phosphate-buffered saline three times to remove all traces of ethanol and ethanol byproducts. The in vitro cytotoxicity study was performed using the ISO 10993-5 extract method on MC3T3-E1 osteoprogenitor cells (ATCC, USA) [52]. Coupon-leached media was produced by soaking the three coupon groups (N = 2) in complete αMEM (Thermofisher, Waltham, MA, USA) media supplemented with 10% FBS and 1% penicillin-streptomycin for 72 h according to a weight ratio of 0.2 g/mL [53,54]. The resulting media and media-leached byproducts were separated into four increasing dilutions (1×, 2.5×, 6×, and 10×). In total, 5000 MC3T3 cells were seeded in a 96-well plate in addition to a calibration curve of increasing cell numbers and were incubated for 24 h at 37 °C and 5% CO_2_. Attachment and seeding density were confirmed with the PrestoBlue Viability Assay (Thermofisher, USA) adhering to the protocol provided by the manufacturer, and increasing dilutions of coupon-leached αMEM were added to the cells. The seeded cells were cultured in the effluent-leached media along with controls grown with complete αMEM. The cell numbers were recorded with the PrestoBlue assay after culturing for 72 h by measuring fluorescence with an excitation wavelength of 530 with a bandwidth of 25 nm and a fluorescence wavelength of 590 with a bandwidth of 35 nm. Percent viabilities were calculated by dividing the final number of cells by the initial number and were normalized by dividing the resulting viability by the average percent viability of the control. Viabilities were imported into R where a two-way ANOVA test using a 95% confidence interval and a post hoc Dunnett test was performed on the resulting data. Weight loss post-incubation in media was measured to determine differences between interactions of nanocomposites and ultra-pure Mg in media.

### 2.7. Microstructure and Corrosion Surface Investigation

A microstructural analysis of the prepared nanocomposites was performed to determine the effect of the BN addition on the microstructure. Prior to SEM analysis, non-corroded samples were polished to produce a mirror surface using a metallographic polishing and grinding machine with sandpapers of reducing grit sizes, etched in an acetic glycol solution composed of 20% acetic acid, 1% nitric acid, and 60% ethylene glycol for 3–5 s, and ultrasonically cleaned for 3 min immediately before SEM analysis [55].

The surface of the prepared samples was characterized and imaged using scanning electron microscopy (SEM) and energy dispersive X-ray spectroscopy (EDX). Samples were prepared according to the sonication steps used prior to cytotoxicity, as previously mentioned. SEM images of the pure Mg samples and immersion samples were taken at magnifications of 500× to visualize the morphology of corroded and non-corroded samples. Separate samples were immersed in complete αMEM media for 72 h to determine corrosion byproducts under the conditions through which leached media was generated in the cytotoxicity studies. Point EDX spectra were detected to analyze the elemental composition of corrosion byproducts that formed on the surface of the samples and the surface composition of ultrasonically cleaned and etched samples. Data were collected at three points on the surface of coupons and the surface was scanned to analyze the overall composition. X-ray diffraction (XRD) was performed at a scan rate of 0.1°/s for samples before and after corrosion in αMEM media for 72 h. XRD diffractograms were developed through graphing detected intensity versus the 2 theta (°) angle and comparing the resulting data to known diffractograms using the X’Pert HighScore software’s built-in search-match algorithm containing elements detected in the previous EDX spectra analyses. Compounds with the highest score based on the search-match algorithm with background signal eliminated were accepted as candidates for corrosive byproducts of nanocomposite and pure Mg corrosion in media.

### 2.8. Statistical Methods

Statistical significance between multiple groups was analyzed with either ANOVA tests if normality was established or a Kruskal–Wallis test due to a lack of data demonstrating normal distribution, as determined in a Shapiro–Wilk test. To determine differences between each group, a Dunnett post hoc was used for ANOVAs and a Dunn post hoc was performed if the Kruskal–Wallis was significant. A confidence interval of 95% was used to confirm significance. In the notation used in the figures, ns, *, **, ***, and **** represent no significance, *p* ≤ 0.05, *p* ≤ 0.01, *p* ≤ 0.001, and *p* ≤ 0.0001, respectively.

## 3. Results

### 3.1. Microstructure Investigation

Microstructure analyses were performed on cleaned and polished samples after etching in acetic glycol using SEM imaging. Samples imaged at 250× were taken immediately post-etching. Figure 1 shows the SEM micrographs of pure Mg and Mg–BN nanocomposite materials investigated in this study. The images show the expected severe distortion of grains and grain boundaries as a result of the hot extrusion process employed in the manufacturing route of the nanocomposites in this study.

### 3.2. Microhardness, Compression, and Porosity

The microhardness of the prepared Mg–BN nanocomposite and pure Mg samples was assessed. The average and standard deviation values obtained from the Vickers microhardness test were 40.1 ± 1.4, 41.1 ± 1.3, and 47.62 ± 1.6 HV for the pure Mg (Mg–0%BN), Mg–0.5%BN, and Mg–1.5%BN vol.% nanocomposite samples, respectively (Figure 2A). It can be observed that the addition of only 0.5 vol.% of the BN nanoparticles did not result in a significant improvement in the mechanical properties, and the addition of a higher content (i.e., 1.5 vol.%) seems to be necessary to notice a significant increase in the microhardness. This represents an approximately 1.2 times increase in the microhardness after the addition of 1.5 vol.% of the BN nanoparticles. The enhancement in the mechanical properties after the addition of the BN nanoparticles can be attributed to the presence of the hard BN nanoparticles along the interfaces of the magnesium matrix grains, which limited the deformation of the matrix grains caused by dislocation movement and twining [33,34,39]. To further investigate mechanical properties, a compression test was performed. Stress–strain curves generated from compression tests on 0–1.5%BN coupons showed a similar behavior to that observed from the microhardness test (Figure 2B), and from these data we calculated offset yield strengths, ultimate strengths, and strains at maximum stresses. Based on the stress–strain curves (Figure 2B), the 1.5%BN nanocomposite samples had much higher yields and ultimate strengths compared to the pure Mg and the 0.5%BN samples. For example, the yield strength increased from 56.8 MPa for pure Mg (0%BN) to reach 69.59 MPa for the 1.5%BN nanocomposite. This also represents an approximately 1.2 times increase in the yield strength after the addition of 1.5 vol.% of the BN nanoparticles.

The density of the nanocomposites and pure Mg samples was measured using a MXBAOHENG MH-300A density meter, which calculates density through Archimedes buoyancy principle. The densities obtained from the density meter were converted to porosity by computing the ratio of measured density to theoretical density and subtracting that ratio from one. The average relative density and density decreased in the line plot from Mg–0%BN to Mg–1.5%BN, as shown in Figure 3A. Therefore, the porosity derived from the density values revealed an increase in porosity with higher vol.% BN addition (Figure 3B). However, the highest calculated porosity level (1.74%), in the case of the 1.5%BN nanocomposites, was within the expected ranges for composite materials produced by the powder metallurgy manufacturing process and it did not result in a deterioration in the mechanical properties. It is also worth mentioning that there was no statistical significance detected between all samples due to the wide range of measurements.

### 3.3. Contact Angle

The hydrophilicity of the nanocomposites was assessed by obtaining contact angle measurements for 6 drops on each side of the nanocomposite or pure Mg coupon, and a total of 12 drops were averaged for each sample. The sessile drop technique was used and, using a light source, we projected the image of the drop on a backboard containing a goniometer to determine contact angle. The hydrophilicity assessment using the contact angle test showed an average contact angle of 125.9 ± 11.2°, 107.6 ± 16.1°, and 124.4 ± 10.7°, for the pure Mg, Mg–0.5%BN, and Mg–1.5%BN samples, respectively, with distilled water on non-sanded surfaces (see Figure 4). An ANOVA test revealed a significant decrease in contact angle for Mg–0.5%BN (Figure 4). However, coupons sanded immediately prior to contact angle measurements showed an average contact angle of 43.7 ± 11.9°, 45.6 ± 8.4°, and 51.9 ± 9.2° for the pure Mg, Mg–0.5%BN, Mg–1.5%BN, respectively, for distilled water (Figure 4). Contact angles measured with simulated body fluid (SBF) demonstrated a similar pattern to distilled water. Without sanding the samples, the average contact angles were 107.4 ± 23.9°, 110.2 ± 20.5°, and 113.2 ± 30.0° for the pure Mg, Mg–0.5%BN, Mg–1.5%BN, respectively, while sanded samples with SBF had average contact angles of 71.9 ± 20.3°, 62.8 ± 17.1°, and 72.8 ± 21.1° for the pure Mg, Mg–0.5%BN, Mg–1.5%BN, respectively. While hydrophilicity tests were repeated with SBF to assess the hydrophilicity more accurately in a physiologically mimetic environment, no statistical significance was detected between each sample group (Figure 3). The wettability of surfaces >90° were hydrophobic materials, and all materials were hydrophobic prior to sanding and hydrophilic post-sanding [56]. The reaction of Mg with air resulted in a MgO surface coat that could contribute to this behavior. Further, the sanded wettability agreed more with the published value for Mg with water.

### 3.4. In Vitro Electrochemical Corrosion

The corrosion current densities of the pure Mg (Mg–0%BN), Mg–0.5%BN, and Mg–1.5%BN samples were obtained by extrapolating the anodic and cathodic curves generated by potentiodynamic polarization (PDP) tests. Corrosion current densities based on the Tafel extrapolation method can be used as an indicator for the corrosion rates and a benchmark of the corrosion resistance. Higher current densities generally mean higher corrosion rates and, hence, less corrosion resistance. Corrosion current densities determined through electrochemical testing, see Table 1, demonstrated that nanocomposites statistically significantly corroded at a decreased rate compared with pure Mg and Mg–0%BN (Figure 5). Interestingly, the Mg–0.5%BN nanocomposite had a significantly lower current density and more positive corrosion potential compared with the Mg–1.5%BN one. The phase shift in corrosion density and corrosion potential (voltage) can be observed in the PDP curves (Figure 5).

The properties of Mg–BN nanocomposites are summarized in Table 1. The addition of BN nanoparticles to a Mg matrix conferred a significant increase in microhardness, as anticipated by previously demonstrated nanocomposite material properties [2,3,4,5,57,58,59]. The hydrophilicity, illustrated by a low contact angle, indicates that optimal anchorage to a bone substrate and cell adhesion was possible [60]. A marginal increase in corrosion resistance compared with that of pure Mg (0%BN) was also predicted to improve the retention of the Mg–BN nanocomposite in a physiological environment.

### 3.5. Cytotoxicity Test

To determine the cytocompatibility of the nanocomposites compared with ultra-pure Mg, we assessed the toxicity of each material on mouse pre-osteoblast cells. According to the ISO 10993-5 standard, if the reduction in cell viability is greater than 30%, the material is considered cytotoxic. The cytotoxicity experiments generally demonstrated an increasing trend of viability from 1× to 10× dilution points (Figure 6A). None of the data points at 1× and 2.5× dilution factors were above or at 70% viability, and the viability increased significantly past 2.5×. All coupon groups at 6× or 10× dilution factors were above 70% on average, with at most one sample out of six measurements having cytotoxic effects with a less than 70% viability. There was no statistically significant difference between the cytotoxicity of the ultra-pure Mg, Mg–0.5%BN, and Mg–1.5%BN Mg groups (*p* > 0.05), suggesting that BN at low vol.% does not significantly impact the cytotoxicity of Mg (Figure 6A).

In addition to cytotoxicity, the rate at which the nanocomposite dissolves in the media was investigated to determine the integrity of the material in near-physiological environments. The sample weight gain/loss of the samples after 72 h did not follow a particular trend (Figure 6B). All ultra-pure Mg samples had a weight gain that ranged from 0.20 to 0.97% with an average of 0.53 ± 0.26%. The Mg–1.5%BN and Mg–0.5%BN coupons contained samples that gained and lost weight after 72 h in αMEM media. The Mg–1.5%BN nanocomposite samples had a weight gain ranging from 0.34 to 1.95%, while the weight of other coupons ranged from 0.035 to 11.26% weight loss. The Mg–0.5%BN nanocomposites, however, varied less in terms of magnitude with weight gains ranging from 2.39 to 4.08% and losses ranging from 0.24 to 0.57%. There was a statistically significant difference between Mg–0.5%BN and ultra-pure Mg, respectively, compared with Mg–1.5%BN. It is worth mentioning that this is a short-term immersion test and it does not produce conclusive results.

### 3.6. Corroded Surface Characteristics

Surface scans using SEM were performed at 500× to characterize the surface morphology and determine the elemental composition of the corroded samples. Samples were ultrasonically cleaned according to standard sterilization protocols and contained some impurities on the surface, which visually increased from ultra-pure Mg to Mg–0.5%BN and Mg–1.5%BN samples (Figure 7A–C). Due to the reactivity of Mg, a high level of surface oxidation was expected. A crystalline material self-assembled during corrosion, which was previously demonstrated in media during media leaching of nanocomposites for cytotoxicity tests (Figure 7D–F). Crystalline corrosive byproducts were observed on all nanocomposites and ultra-pure Mg samples with severely high levels of crystallinity were observed in the Mg–1.5%BN samples. 

To better understand the surface byproducts observed after corrosion in media, surface EDX was performed on each sample and assessed for elemental composition. The amount of oxygen and carbon detected increased in all corroded samples compared with their ultrasonically cleaned counterparts (Figure 8). Furthermore, the oxygen and carbon accumulation were higher in both nanocomposites compared with ultra-pure Mg at all instances (Figure 8). These data support the increased rate of corrosion in Mg supplemented with boron nitride compared with ultra-pure Mg samples observed in both immersion and electrochemical corrosion tests. Additional peaks were detected for chloride and sodium, which could be a byproduct of the media composition. The α-MEM media contained organic salts, essential nucleosides, fetal bovine serum, and a sodium bicarbonate buffer, which could contribute to the EDX results. At higher levels of magnification, the detection of boron and nitrogen could not be achieved, which could be attributed to the low vol.% of BN or any signal from boron and nitrogen being masked by the high levels of oxidation byproducts (Figure 8B,D).

Using the elements detected in the EDX analyses, a search-match algorithm was used restricting compound candidates to the elements in Figure 8 for the XRD diffractogram collected for ultra-pure Mg, Mg–0.5%BN, and Mg–1.5%BN. The top-scoring candidate using the X’Pert HighScore Plus software’s bult-in search-match algorithm for ultrasonically cleaned ultra-pure Mg compounds was magnesium peroxide (Figure 9A), which was also the top scoring candidate for ultrasonically cleaned Mg–0.5%BN and Mg–1.5%BN (Figure 9B,C). These data indicate that the sterilized samples exposed to air will still oxidize, prevalently producing magnesium peroxide, which was expected due to the reaction of magnesium with air. However, the top-scoring candidates in corroded ultra-pure samples were magnesium oxide in addition to magnesium peroxide. Magnesium oxide was also detected in corroded Mg–BN0.5 samples, but was absent in corroded Mg–BN1.5. Interestingly, corroded Mg–BN0.5 and Mg–BN1.5 had boron nitride (BN) as a top candidate despite virtually zero detection using EDX. Detection indicated that a small level of BN aggregation had occurred during corrosion in media for both vol.% of Mg–BN nanocomposites. Magnesium oxide post-corrosion was only detected in the Mg–0.5%BN nanocomposite and ultra-pure Mg, which confirms an accelerated mass-loss of corrosion byproducts with higher concentrations of BN incorporated in the Mg matrix. However, further validation with more formulations of different vol.% nanocomposites is needed.

## 4. Discussion

We synthesized and characterized a novel Mg-based nanocomposite using Mg powder and BN nanoparticles. To investigate whether the addition of BN altered the microstructure of the material, we examined polished and etched samples using SEM. Microstructural analyses suggested a slight grain refinement associated with the increased vol.% of the BN nanoparticles (Figure 1A). However, a lack of statistical significance failed to provide a conclusive difference between Mg with and without BN.

The mechanical strength of pure Mg is insufficient to support the constant stresses applied on orthopedic implants in vivo. It has been previously demonstrated that the addition of BN nanoparticles to Mg matrices results in improved creep resistance [61]. The shoulder-like behavior that can be observed in the stress–strain curves around yielding is the “yield plateau” phenomenon, which occurs during the compressive deformation of some Mg alloys. This yield plateau is associated with the onset of twinning during loading along the extrusion direction of the test specimen and it represents a region of large twin volume fraction nucleating and propagating [62]. The microhardness results in this work support this observation with statistically significant increases in microhardness between the nanocomposites and pure Mg groups. Using an ANOVA test and a Dunnett post hoc comparison, each group demonstrated statistical significance, supporting the trend that increased vol.% of nanoparticles corresponds to an increase in microhardness and, hence, mechanical strength (Figure 1A). Compression tests further corroborated the microhardness results, as the ultimate strength increased with increasing vol.% of BN added to Mg (Figure 1B). The presence of hard BN nanoparticles along the interfaces of the Mg matrix increased the strength by hindering mobile dislocations and twinning, which increased the hardness [63,64]. The strengthening of the Mg–BN nanocomposites can also be attributed to BN nanoparticles preventing crystal growth and, therefore, playing a role in refining the grain matrix [65,66,67,68,69].

Hydrophilicity significantly impacts the cell attachment to the biomaterial, which influences the integration of the material with bone [56,70]. The published value of pure Mg is ~32.2° in water, which is hydrophilic and at a suitable range for cell adhesion. We performed contact angle tests using both distilled water and simulated body fluid (SBF) to compare against other similar nanocomposites, which are typically measured with water, and provided a more physiologically relevant standard. Contact angles for all surfaces without sanding before analysis were highly hydrophobic with all angles measuring >90°, which is the established threshold for a material to be hydrophobic (Figure 3). Experiments replicated for freshly sanded surfaces immediately prior to analysis, however, revealed hydrophilic surfaces with average contact angles all being <90° in pure Mg, Mg–0.5%BN, and Mg–1.5%BN (Figure 3). The contact angle for most orthopedic implants range between 58.5 and 81.7° in distilled water [71]; therefore, the measured contact angles of the Mg–BN nanocomposites in this work are in range of the desirable hydrophilicity.

Enhancing corrosion resistance is critical for Mg due to the fast degradation rate of pure Mg, limiting its use as an orthopedic implant. Using the nanocomposites synthesized in this study, we detected a decrease in the corrosion current density, determined based on the PDP Tafel curves, between Mg–0%BN, Mg–0.5%BN, and Mg–1.5%BN. Corrosion current densities based on the Tafel extrapolation method can be used as indicators for the corrosion rates and a benchmark of the corrosion resistance. Interestingly, corrosion density also significantly increased between Mg–0.5%BN and Mg–1.5%BN, indicating that the electrochemical corrosion is faster in higher vol.% BN but slower in lower vol.% BN. In contrast, biocompatibility immersion testing indicated that the mass loss increased in nanocomposites compared with the ultra-pure Mg control group. However, this difference in corrosion resistance can be attributed to the differences in the purity levels and the manufacturing method used to make the nanocomposites (powder metallurgy) and the ultra-pure control (hot-rolling). These data suggest that increased microhardness and strength is not entirely indicative of increased corrosion resistance. It has been previously demonstrated that a nickel–tungsten alloy supplemented with increasing concentrations of BN nanoparticles showed a decreased corrosion resistance in electrochemical tests followed by a decrease in resistance upon further increases in BN [72].

The results of the biocompatibility studies measuring the cytotoxicity of nanocomposite and ultra-pure Mg-leached media demonstrate that BN nanoparticle addition does not confer cytotoxic effects. At the dilutions of 6× and 10×, the average% viabilities were above 70% normalized viability compared with non-treatment. This indicates that 70% of cells were retained post-treatment with leached media for 24 h in all groups. Because there was no statistical significance between nanocomposite and ultra-pure Mg cytotoxicity, we can conclude that BN nanoparticles do not confer cytotoxic effects, but also do not enhance the cytocompatibility. Long-term assessments of viability overtime, however, need to be investigated to further support this conclusion. In vitro coupon weight-loss in media after 72 h suggested that longer corrosion times are required to determine the corrosive variation between Mg–0.5%BN, Mg–1.5%BN, and ultra-pure Mg due to inconsistent results. This assessment of weight loss to estimate corrosion rate may be inaccurate during short corrosion periods, as these data are inconsistent with SBF corrosion demonstrated in the literature. These data also indicate that surface coatings enhancing cytocompatibility and increasing corrosion resistance are necessary for in vivo applications.

Surface characterization post-corrosion in media showed that nanocomposites appear to oxidize faster than ultra-pure Mg in ultrasonically cleaned samples based on byproducts appearing on SEM images of nanocomposites alone. Furthermore, carbon and oxygen content significantly decreased in corroded nanocomposites based on EDX spectra results. Further investigation with XRD revealed that this oxygenation can be attributed to magnesium oxide (1.5 BN) and magnesium peroxide (0.5 BN) present on the surface. However, there is little explanation for the increase in carbon content observed in the EDX analyses.

## 5. Conclusions

The use of low contents (0.5 vol.% and 1.5 vol.%) of BN particles to strengthen the Mg matrix was examined for mechanical properties, corrosion behavior, and cytotoxic effects for skeletal fixation hardware applications. In terms of the distortion of grains and grain boundaries due to the hot extrusion process, the microstructure of Mg–BN nanocomposites does not appear to be impacted by the addition of BN compared to pure Mg. The addition of nanocomposite BN particles, however, increased the strength and microhardness of the Mg matrix by limiting the dislocation movement and twining, which is critical due to the low strength of pure Mg. The addition of BN also increased the corrosion resistance in electrochemical tests compared with 0% BN–Mg synthesized in a similar manner. The Mg–BN nanocomposite samples showed similar hydrophilic contact angles with SBF and in vitro cytotoxicity levels to those for the ultra-pure Mg when coupons leached in α-MEM media were exposed to MC3T3-E1 cells. The EDX and XRD data revealed the identity of the byproducts of corrosion in media to be primarily oxidated Mg with a more crystalline appearance in the Mg–BN nanocomposite samples. The impact of this behavior requires further investigation to determine the nature of this reaction in media. Collectively, these data suggest that the strengthening of the Mg matrix improved electrochemical corrosion behavior. The contact angle analysis indicates that surface modification (e.g., coatings) is essential to protect the Mg from surface oxidation, which dramatically increases hydrophobicity based on contact angle. Further, coatings to reduce immersion corrosion rates are also required to retain implant function for orthopedic applications. Further investigation of the interaction of BN nanocomposites with a biological environment and improvements in synthesis methods and surface modifications is needed.

## Figures and Tables

**Figure 1 bioengineering-10-00757-f001:**
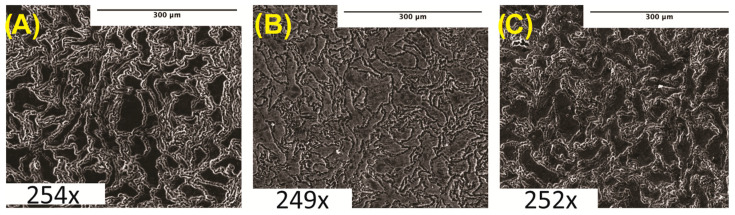
**Microstructure analysis:** (**A**) pure Mg (0% BN) SEM image. (**B**) Mg–0.5%BN SEM image. (**C**) Mg–1.5%BN SEM image. Magnifications are included at the bottom of each image.

**Figure 2 bioengineering-10-00757-f002:**
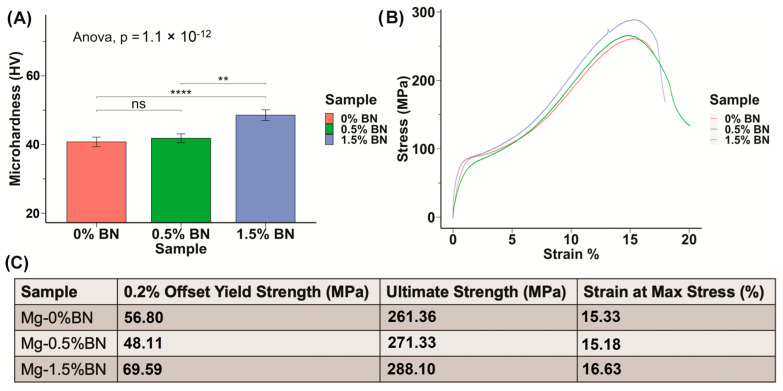
**Mechanical Properties:** (**A**) The microhardness of sample determined through the Vickers microhardness test demonstrates an increasing trend in hardness with increased vol.% of BN nanoparticles included in the Mg matrix. (**B**,**C**) Stress–strain curves derived from compression test results for nanocomposites compared with 0% BN–Mg illustrates an increased strength with higher vol.% BN addition (ns, ** and **** represent no significance, *p* ≤ 0.01 and *p* ≤ 0.0001, respectively).

**Figure 3 bioengineering-10-00757-f003:**
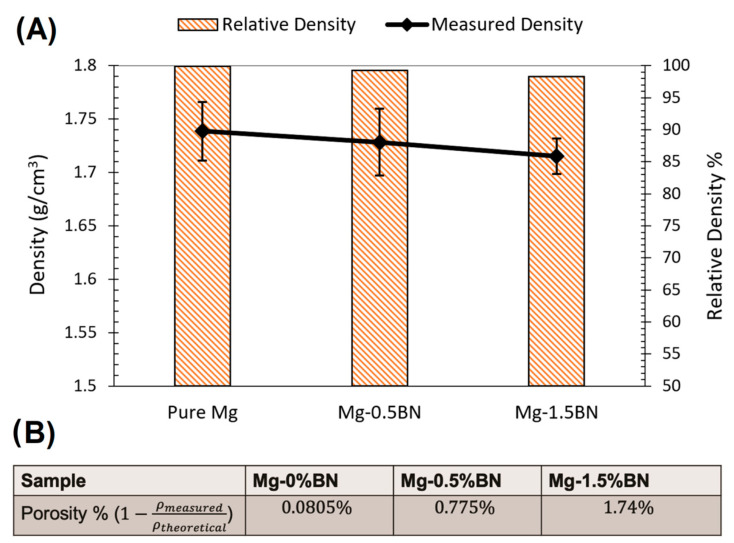
**Density and Porosity:** (**A**) Relative density and density measurements are represented as a bar plot and line plot, respectively. A decrease in density corresponded to a decrease in relative density with higher vol.% BN addition. (**B**) The porosity was calculated from the average of the density measurements divided by the theoretical density. With higher vol.% BN addition, the porosity increased.

**Figure 4 bioengineering-10-00757-f004:**
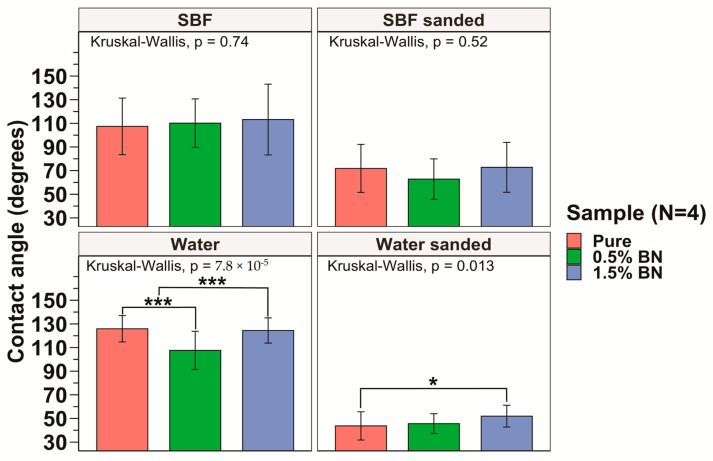
**Contact Angle Measurements:** The measured contact angles for the pure Mg, Mg–0.5%BN, and Mg–1.5%BN nanocomposite samples demonstrated that pure Mg and Mg–1.5%BN had statistically significantly higher contact angles compared with those of Mg–0.5%BN. There was no significant difference between contact angles for SBF (* and *** represent *p* ≤ 0.05 and *p* ≤ 0.001 respectively).

**Figure 5 bioengineering-10-00757-f005:**
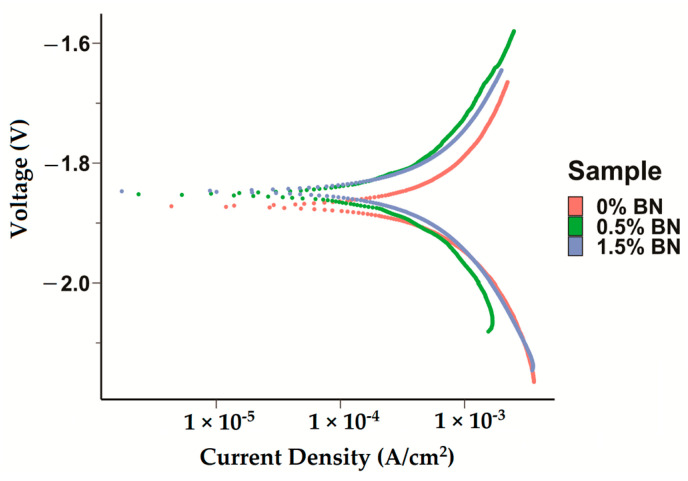
**Electrochemical Corrosion Tests:** Potentiodyanmic curves for the pure Mg, Mg–0.5%BN, and Mg–1.5%BN nanocomposite samples generated by applying a Tafel fit using the Gamry interface was detected, with phase shifts decreasing voltage in nanocomposites.

**Figure 6 bioengineering-10-00757-f006:**
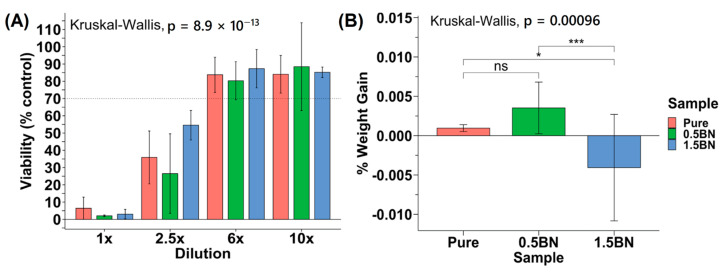
**Cytotoxicity and in vitro corrosion:** (**A**) Average percent viability of control seeding dilution points compared across Mg coupon compositions (N = 2) for 5000 MC3T3 cells seeded in a 96-well plate. (**B**) Percent weight gain/loss after 72 h in media SBF (ns, *, and *** represent no significance, *p* ≤ 0.05 and *p* ≤ 0.001, respectively).

**Figure 7 bioengineering-10-00757-f007:**
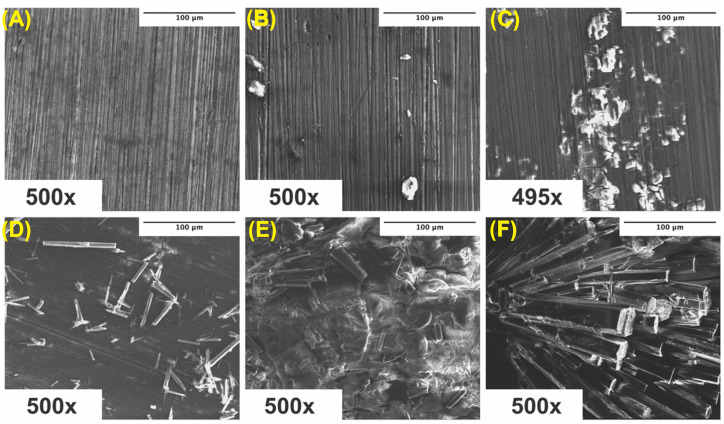
**SEM Images Pre- and Post-Corrosion:** SEM images of ultra-pure Mg (**A**), Mg–0.5%BN (**B**), and Mg–1.5%BN (**C**) coupon surfaces at depths of 500× after ultrasonic and UV sterilization for cytotoxicity testing, SEM images of ultra-pure Mg (**D**), Mg–0.5%BN (**E**), and Mg–1.5%BN (**F**) after 72 h corrosion in α-MEM media at 37 °C and 5% CO_2_ for media-leaching cytotoxicity tests transferred to an SEM for analysis.

**Figure 8 bioengineering-10-00757-f008:**
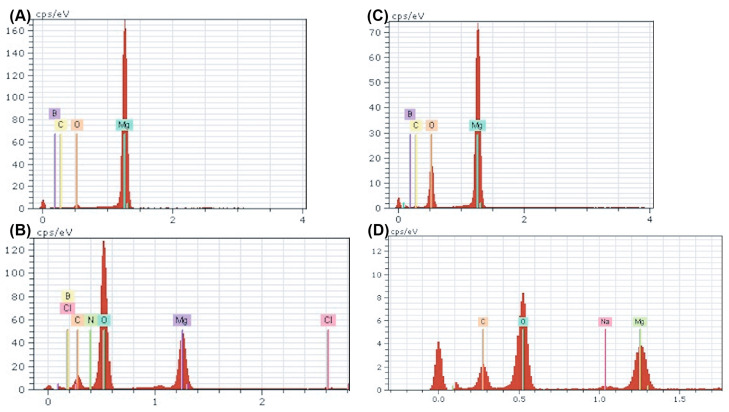
**EDX Spectra of Nanocomposites Pre- and Post-corrosion:** Energy dispersive X-ray point spectra for Mg–BN0.5 (**A**,**B**) and Mg–BN1.5 (**C**,**D**) before and after corrosion in αMEM media for 72 h. An increase in oxygen and carbon content was detected in both nanocomposites post-corrosion in addition to trace amounts of potential salt deposits.

**Figure 9 bioengineering-10-00757-f009:**
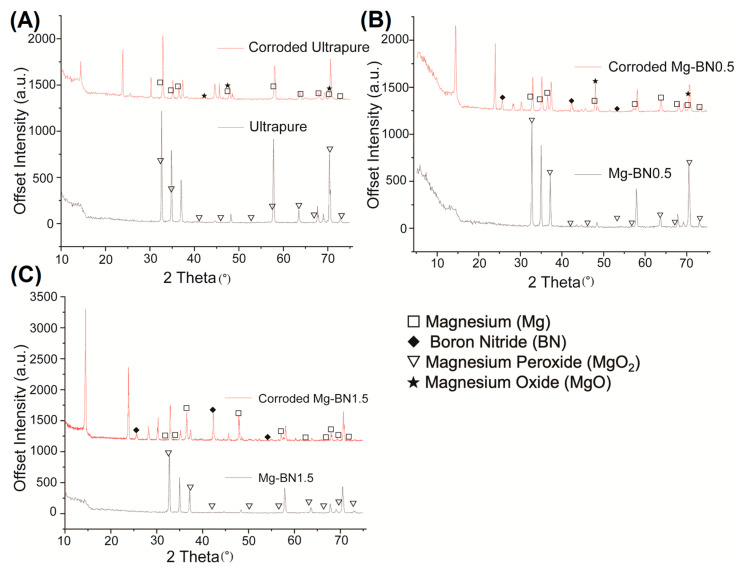
**XRD Diffractogram Pre- and Post-corrosion:** X-ray diffraction diffractogram for ultra-pure Mg (**A**), Mg–0.5%BN (**B**), and Mg–1.5%BN (**C**) before and after corrosion in αMEM media for 72 h. Corroded 1.5 BN nanocomposite samples resulted in a high scoring detection of boron nitride and magnesium. The corroded 0.5 BN samples contained the same byproducts as corroded 1.5 BN in addition to magnesium oxide. Non-corroded samples still had magnesium peroxide detected on the surface of all samples, including ultra-pure Mg.

**Table 1 bioengineering-10-00757-t001:** **Properties Summary:** Summary of the main characteristics of pure Mg (0%BN), Mg–0.5%BN, and Mg–1.5%BN nanocomposites.

Sample	Microhardness (HV)	Ultimate Strength (MPa)	Contact Angle SBF Sanded (°)	Corrosion Potential (V)	Corrosion Current Density (μA/cm^2^)
Mg–0%BN	39.98 ± 1.36	261.3	71.9 ± 20.3°	−1.87	770
Mg–0.5%BN	41.02 ± 1.24	271.3	62.8 ± 17.1°	−1.85	417
Mg–1.5%BN	47.62 ± 1.53	288.1	72.8 ± 21.1°	−1.85	558

## Data Availability

Not applicable.

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
