# Peer review of "Characterization and Biocompatibility Assessment of Boron Nitride Magnesium Nanocomposites for Orthopedic Applications"

_bioengineering, 2023, doi:10.3390/bioengineering10070757_

Round 1
Reviewer 1 Report
The paper investigates on the mechanical properties, corrosivity, and cytotoxicity of Mg nanocomposites reinforced with boron nitride (BN) nanoparticles (0.5 and 1.5 vol % BN) for orthopaedic fixation device applications. Micro-hardness, compression tests, in vitro corrosion, contact angle, and cytotoxicity tests have been also performed on in vitro specimens to investigate the effect of the BN nano-particles on the mechanical properties. Biocompatibility assessments showed that the incorporation of BN nanoparticles had no significant impact on the cytotoxicity of Mg. The obtained results seem to confirm that the addition of BN nanoparticles to the Mg matrix can increase strength and corrosion resistance without influencing cytotoxicity in vitro. Indeed, the authors rightly declare that further investigations of the interaction of BN nanocomposites with a biological environment and improvements in synthesis methods and surface modifications is needed.
The background to the problem is sufficiently covered and the paper is well structured giving a clear explanation of the work carried on.
In my opinion the paper can be accepted for publication.
The quality of Englsh is ok
Reviewer 2 Report
An interesting nanocomposite was characterized. The applied methods are relevant and the results are important.
Minor corrections are needed:
1. it would have been better to test the hydrophilicity of the samples with dynamic contact angle measurements.
2. Chapter 3.1., last sentence is more an assumption, please rephrase.
3. Chapter 3.2. what are representing the values beside the average values: SD or SEM?
4. Figure 2, in the table values should be given with the same nr. of digits
5. In every bar graph, if you give an average value (for e.g. Figure 6), please give the standard error of the mean instead of the standard deviation, as that is giving the statistically relevant information
Reviewer 3 Report
The present study introduces a very interesting investigation regarding the intruding BN nanoparticles into Mg-based alloys. The work shows the positive effect of Mg-based nanocomposites on some properties, such as increase strength and corrosion resistance without influencing cytotoxicity in vitro. However, for this manuscript to become publishable, some points must be considered and reviewed by the authors, which are explained below.
Fig. 9: The phenomenon of X-ray diffraction does not generate spectra. Here you actually have diffractograms, method, which record diffraction patterns. Please correct this conceptual error.
Fig. 1. What is the meaning of numbers 254X, 249X, 252X? For my opinion just the scale will be enough.
Fig. 2b: Can the authors explain the reason of appearance of a shoulder in the Stress-strain curves at 2% approximately?
Page 6 line 283: “…approx.” no need to use abbreviations.
Fig. 2a: what is the meaning of “Anova” ?
Fig. 8 has very bad quality; the authors should improve it.
Some correction in English shouls be done.
